# Study on Sports, Extracurricular Activities, Electronic Device Usage Factors Associated with Chronic Fatigue Syndrome in Taiwanese Preschoolers

**DOI:** 10.3390/children10081278

**Published:** 2023-07-25

**Authors:** Su-Fen Huang, Hui-Ying Duan

**Affiliations:** 1Department of Early Childhood Education, National Taitung University, No. 369, Sec. 2 University Road, Taitung City 950309, Taiwan; kou@nttu.edu.tw; 2Department of Infant and Child Care, National Taipei University of Nursing and Health Sciences, No. 365 Min-Te Road, Taipei 112303, Taiwan

**Keywords:** fatigue syndrome, sports, extracurricular activities, electronic device usage

## Abstract

Under the impact of the pandemic, electronic device usage has become the primary tool for learning. Due to social distancing restrictions, many sports facilities have been forced to close, resulting in changes in daily activities for preschool children. This research aimed to investigate the sports, extracurricular activities, and electronic device usage factors associated with chronic fatigue syndrome among Taiwanese preschoolers. Five-year-old preschoolers were randomly selected using a stratified multi-stage random cluster sampling method. The parents of the preschoolers completed the questionnaires, which contained items related to the symptoms of fatigue, extracurricular activities, and electronic device usage of their preschoolers. A total of 1536 valid questionnaires were returned. The data were then analyzed using descriptive statistics and the chi-square test. The following results were obtained: (1) the preschoolers who exercised at least three times per week, engaged in sweating exercise for at least 30 min at a time, had a continuous rhythmic exercise habit, and participated in a variety of exercise types experienced a lower degree of fatigue; (2) the preschoolers who engaged in extracurricular activities every day exhibited a higher degree of fatigue; (3) the preschoolers who watched television or used smartphones to pass the time due to boredom, watched television or used smartphones on holidays, played video games or surfed the Internet due to boredom, and played video games or surfed the Internet on holidays displayed a higher degree of fatigue. This research verified that regular exercise with various sports, extreme extracurricular activities, and laissez-faire electronic device usage are factors associated with fatigue syndrome in preschoolers. It is suggested to develop children’s regular exercise habits, avoid excessive extracurricular activities, and guide their electronic device usage.

## 1. Introduction

Regulations on preschool education in Taiwan require preschools to provide more than 30 min of gross motor activity for sweating every day [1]. The Early Childhood Education and Care Curriculum Framework also points out that the goal of healthy physical movement is to ensure that children like sports [2]. Article 43 of “The Protection of Children and Youths Welfare and Rights Act” indicates that children and youth shall not continue using electronic products for an unreasonable amount of time, causing harm to their physical and mental health. Article 91 also mentions that “parents, guardians or other people looking after children and youth who violate the regulations described in Article 43 will be fined.” [3]. It is apparent that preschool education in Taiwan attaches great importance to preschoolers’ sports to promote their physical and mental health while avoiding the adverse effects of prolonged use of electronic devices.

Under the impact of COVID-19, preschools in Taiwan were intermittently closed or conducted online without many outdoor activities, thus resulting in an increase in the use of electronic devices. When children returned to school, they were less energetic and appeared lethargic [4]. The question lies in whether this phenomenon is due to a lack of exercise at home or because of the use of electronic devices. More attention should be paid to the phenomenon of fatigue and lack of energy in children.

Yu [5] investigated the lifestyle, healthy status, and activity time of 1050 children aged three to six in Taipei City and found that 12.64% of children woke up in the morning with a strong desire to sleep, and the average time of outdoor play was 58 min. Three-year-olds have more indoor and less outdoor play time, and nearly 50% of the children have extracurricular activities. The average TV viewing time for young children is 1 h and 16 min. Even in rural cities such as Chiayi, before the age of two, 37.2% of children started to watch TV, and 22.3% were exposed to electronic screen media other than TV, including computers, tablets, and smartphones. Before reaching the age of six, 71.2% of children used smartphones. On average, five-year-old children used these four electronic screens for 1.4 h per day; these children were moderately dependent on screen use [6]. The study of Taiwanese parents arranging extracurricular activities for preschoolers shows that 51.5% of the children interviewed participated in talent-learning classes, 30% joined drawing classes, and 24% had English lessons. Each child’s average time spent on talent learning was 2.25 h per week [7]. Huang and Maehashi [8] investigated the status and factors of the leisure activities of preschool children and concluded that nearly 94% of preschoolers perform leisure activities at home, and drawing and watching television programs are the most popular leisure activities (50%). These researchers pointed out the importance of leisure activity participation for preschool children’s development and encouraged the involvement of parents to foster positive thoughts about leisure activities for preschool children.

Moscatelli et al. [9] examined the lifestyle, eating habits, and impact of nutritional education on undergraduate students in southern Italy. Educational events can enhance awareness of the health risks associated with unhealthy lifestyles, leading to lifestyle improvements. Jin et al. [10] used the Self-rated Fatigue Scale (SFS) to measure the fatigue status of 726 Chinese students ranging in age from 13 to 26 years old. The risk factors for fatigue status include myopia, partial eclipse, food pickiness, lack of sleep, and drinking. Eddy and Cruz [11] reviewed and synthesized the literature on fatigue in children with chronic health problems and examined its relationship to quality of life. Their study found that children often reported fatigue, which decreased their quality of life. 

According to the diagnostic criteria for CFS in children initially published by the International Association for Chronic Fatigue Syndrome in 2005 and republished in 2006, the seven primary symptoms are fatigue, sleep dysfunction, pain (e.g., headaches and muscle and abdominal pain), and neurocognitive, autonomic, neuroendocrine, and immune manifestations [12]. In addition to the aforementioned symptoms, CFS in children is also associated with numerous psychological and behavioral problems, such as mood disturbance, cognitive dysfunction, memory problems, and concentration problems [13,14,15]. Moreover, Richards [16] stated that, despite the similarity of the presentation of CFS in children and adults, the diagnosis of CFS varies between the two groups; the physical and psychological symptoms and symptom combinations vary among children and are easily misdiagnosed by pediatricians as manifestations of aleinophobia or emotional disorders. The literature has indicated that the determination and prediction of CFS in children are challenging. Accordingly, understanding the causes and improving the timely prediction of CFS in children, as well as treating and preventing such symptoms, are topics that warrant in-depth investigation.

Solomon-Moore et al. [17] pointed out 25 years ago that children and young people with CFS were assumed to be relatively physically inactive. Although there is no evidence supporting their claim, it is not defined in all cases. Light activity was associated with better physical functioning and lower fatigue than inactivity. Japanese scholars have suggested that fatigue in young children is significantly associated with staying up late, irregular outdoor sports, dietary habits, skipping breakfast, late-night snacks, excessive extracurricular activities, and prolonged exposure to electronic devices [18,19,20,21]. 

In summary, outdoor sports, extracurricular activities, and electronic device usage may disturb the rhythm of life and result in fatigue symptoms in young children. However, are these variables associated with the fatigue symptoms of Taiwanese preschoolers? In other words, a lack of sports, busy extracurricular activities, and too many electronic screens may be associated with fatigue in children. The present study investigated the actual associations one by one.

Related questionnaires have long been developed internationally to address fatigue symptoms, and scholars in Taiwan believe that they have achieved good performance assessments [22]. However, when the research subjects of studies are preschool children, it must be considered that these children have limited cognitive ability, do not easily understand the meaning of the questions or the text, and are not suitable for self-administered questionnaire assessments. Japanese scholars have indicated that the answers given by their primary caregivers based on the child’s fatigue have good reliability [23]. Considering the applicability of these questionnaire assessments to young children, this study modified a questionnaire from the Japanese scale, as Japan has a similar cultural background to Taiwan. Afterward, parents were asked to fill out the questionnaire based on their children’s condition; their replies were employed as a basis for the children’s fatigue symptoms. 

In addition, based on relevant literature, this study further investigated and analyzed the health habits of young children (such as dietary habits and sleep quality) and daily activities (such as sports, extracurricular activities, and electronic device usage) to determine whether there are significant differences between these factors and children’s fatigue symptoms. This analysis will help parents and teachers identify the children’s symptoms of fatigue early and implement effective strategies to improve these symptoms so that the children can regain their health and vitality as soon as possible. It also raises the attention of society to the issue of CFS in children and addresses the lack of empirical studies in Taiwan.

## 2. Methods

### 2.1. Subjects and Sampling

#### 2.1.1. Enrollees

A stratified multi-stage random cluster sampling technique was employed to select children aged 5 to 6 years attending registered public or private preschools in Taiwan during the 2020–2021 school year. Preschoolers with specific physical conditions, weaknesses, illnesses, or chronic systemic diseases were excluded from the study. A total of 1536 preschoolers, consisting of 777 boys (50.59%) and 759 girls (49.41%), were enrolled in the study.

Regarding the educational background of the parents, the majority held a junior college or college degree (n = 857; 55.79%). Moreover, most parents reported an annual household income ranging from TWD 500,000 to TWD 1,140,000 (n = 813; 52.93%).

The Human Research Ethics Committee of National Cheng Kung University reviewed and approved the study on 22 October 2019. 

#### 2.1.2. Sample Size

For estimating the sample size, Education Statistics [24] reported an estimated total of 194,317 children between the ages of 5 and 6 registered in public or private preschools in Taiwan during the 2020–2021 school year. To calculate the sample size (n = Z^2^ × *p*(1 − *p*)/ε^2^) with a 95% confidence coefficient and a sampling error of 3%, a minimum sample size of 1061 was required.

According to Sudman [25], for nationwide survey research, the recommended average sample size ranges from 1500 to 2500. Considering this recommendation, a sample size of 2000 children was determined for this study.

In Taiwan’s public and private preschools, a senior class typically consists of approximately 20 children. Consequently, the number of preschools selected for sampling (n = 100) was obtained by dividing the total sample size of 2000 children by the average class size of 20. Therefore, approximately 2000 children from 100 preschools were included in the sample for this study.

#### 2.1.3. Stratified Multi-Stage Random Cluster Sampling

Stratified multi-stage random cluster sampling was implemented to ensure representative coverage of Taiwan’s 359 townships, cities, and districts. Thirteen strata were created, which included one stratum based on ethnicity (“Hakka”) and three strata based on geographic location (“mountainous”, “eastern”, and “outlying islands”). The remaining nine strata were derived by dividing Taiwan into northern, central, and southern regions, and within each region, three additional strata were created based on population density.

Within each stratum, a probability proportional to size sampling method was employed to select a sample of public and private preschools. A total of 104 preschools were selected through this process. Subsequently, a simple random sampling technique was utilized to select 20 children from each of the selected preschools. This resulted in a total sample size of 2080 children, representing various regions across Taiwan. 

Participant recruitment followed a random sampling approach, with selected preschools contacted via phone to assess their willingness to participate. The researchers personally explained the study’s purpose, target population, and content to the preschoolers, providing them with an invitation letter and study explanation. Upon confirmation of participation, willing preschools received a formal request. Parents of children aged five or above in the respective classes were given a sealed envelope containing a parent invitation letter and a questionnaire with clear instructions. Parents were requested to complete the questionnaire and return it in a sealed envelope with their child to the preschool teacher. The researcher or an assistant collected the questionnaires. The researchers respected the decisions of preschools, teachers, or parents who declined to assist or complete the questionnaire. To enhance adherence to the intervention, each interested parent received a small gift with the consent form and questionnaire. This gesture aimed to encourage active involvement and commitment, fostering motivation and a sense of value among the participants.

#### 2.1.4. Questionnaire Administration and Retrieval

A comprehensive distribution of consent forms and questionnaires on chronic fatigue syndrome (CFS) in children and associated factors was conducted among a total of 104 public and private preschools. A total of 2080 sets of consent forms and corresponding questionnaires were distributed. Of these, 1820 sets were returned by participants. After carefully screening for invalid responses, a total of 1536 valid questionnaires were retrieved, resulting in a valid return rate of 78.7%.

### 2.2. Measurements

The survey questionnaire utilized in this research consisted of several sections. Firstly, a demographic data section was included to gather relevant information about the participants. Secondly, an assessment scale for chronic fatigue syndrome (CFS) in children was incorporated to evaluate the presence and severity of CFS symptoms. Thirdly, a survey on children’s health-related habits was conducted, focusing on factors such as sleep quality and dietary habits. Lastly, a section on children’s daily activities was included, encompassing inquiries regarding sports involvement, participation in extracurricular activities, and usage of electronic devices.

Due to the limitation of length, children’s health-related habits were analyzed in another report [26]. This study focused on the associations of sports, extracurricular activities, and electronic device usage with fatigue syndrome in young children. Questions on children’s outdoor activities are in Appendix A; questions on children’s extracurricular activities are in Appendix B; and questions on children’s electronic device usage are in Appendix C. 

#### 2.2.1. Assessment Scale for Symptoms of Fatigue in Children

In order to meet the needs of Taiwanese studies and to analyze the research results with international studies, the Chinese version of the assessment scale for CFS in children with the same content was not developed. Instead, this study adopted the assessment scale for CFS in children with similar cultural backgrounds by Japanese scholars Hattori et al. [18]. For this study, the CFS of children was measured based on three dimensions: sleepy and inactive/blunted responses/lacking energy, difficulty concentrating, and localized pain, totaling ten items. Each of the ten items was rated on a five-point Likert scale, with scores of 1, 2, 3, 4, and 5 indicating “never”, “rarely”, “sometimes”, “very often”, and “always”, respectively. The total score ranged from 10 to 50; higher scores indicated that the children exhibited a higher degree of fatigue, whereas lower scores indicated that the children exhibited a lower degree of fatigue. The parents provided answers based on their observations of their children. The scale’s internal consistency and reliability are good, and it is appropriate for assessing the symptoms of fatigue in young children.

#### 2.2.2. Profile of Children’s Daily Activities Questionnaire

This study investigated children’s daily activities by focusing on three dimensions: sports, extracurricular activities, and electronic device usage. The items were referenced from Huang’s questionnaire on outdoor activities for young children [27], Lee and Tu’s [7] questionnaire on extracurricular activities for young children, and Chen’s and Chang et al.’s questionnaires on electronic device usage for young children [28,29]. The four referenced questionnaires were revised, organized, and analyzed in relation to the research focus of this study to formulate items based on a nominal scale. The parents could choose answer options [“almost” (positive) or “hardly” (negative)] according to their observations of their children. 

In order to assess the validity of the questionnaires developed for this study, a rigorous process was followed. Pretest questionnaires were formulated and presented to a panel consisting of five experts and scholars. The panel evaluated the items to ensure their appropriateness and relevance. Their feedback and suggestions were sought regarding the addition, removal, combination, and other aspects of the items.

After the pretest questionnaires were collected, modifications were made to the wording of certain items and the overall structure of the questionnaire based on the panel’s opinions. These adjustments aimed to enhance the clarity of item meanings, improve the flow of phrases, and further refine the questionnaire structure.

To assess the reliability of the questionnaire, a test-retest approach was employed. Random sampling was used to select 200 parents of children from 30 public and private preschools in Taitung County. Participants completed the questionnaire twice, with a four-week interval between the test and retest administrations. The test-retest reliability of the Assessment Scale for CFS in Children and the Children’s Daily Activities Questionnaire (covering sports, extracurricular activities, and electronic device usage) was found to be 0.92 and 0.90, respectively. Both measures reached a statistically significant level (*p* < 0.05).

### 2.3. Data Analyses

For the statistical tests, children’s fatigue level was considered the explanatory variable, and the current status of sports, extracurricular activities, and electronic device usage were considered the response variables. The dependent variables were combined into positive and negative categories for descriptive statistics and the chi-square homogeneity test. If the results of the test were statistically significant or correlated, it would indicate that children’s fatigue symptoms were statistically different in the related factors; otherwise, the opposite would be true. In addition, the total scores were grouped by the extreme cohort method to understand the overall degree of fatigue symptoms in all children [30]. The scores of the fatigue questionnaire were divided into three groups: “high”, “medium”, and “low”. From high to low, the first 27% of the scores were classified as a high degree of fatigue, the last 27% were classified as a low degree of fatigue, and the scores in between were classified as a medium degree of fatigue. When there is a significant difference in the statistical test results, the Bonferroni correction Z Test is performed for post hoc comparison.

## 3. Results

### 3.1. Overview of Assessment of Symptoms of Fatigue in Children

The frequencies and percentages obtained from grouping the total scores of the Assessment Scale for SF in Children using the extreme high/middle/low groups approach were 436 (28.38%), 717 (46.67%), and 38. Table 1 presents the prevalence rates of common fatigue-related symptoms among preschool children. A significant proportion of children, specifically 41.86%, frequently exhibit yawning during the daytime. Moreover, 39.32% of preschoolers frequently experience feelings of tiredness, while 39.13% frequently display signs of drowsiness. 

### 3.2. Analysis of Differences in the Sports of Children with Varying Degrees of Fatigue

Exercise at least three times per week [χ^2^ (2) = 6.05, *p* = 0.049 *p* < 0.05], sweating exercise for at least 30 min at a time [χ^2^ (2) = 11.38, *p* = 0.003 *p* < 0.01], continuous rhythmic exercise habit [χ^2^ (2) = 13.94, *p* = 0.001 *p* < 0.001], and variety of exercise types [χ^2^ (2) = 9.11, *p* = 0.011 *p* < 0.05] were significantly associated with fatigue degrees in young children. Post hoc comparisons revealed that those with significantly lower degrees of fatigue who checked “almost” for the sports option were those who exercised at least three times per week, sweated for at least 30 min each time, continued regular exercise habits, and had a variety of types of exercises. In addition, there was no significant difference between those who took outdoor excursions on holidays [χ^2^ (2) = 5.71 *p* = 0.058 > 0.05], played outdoors after school [χ^2^ (2) = 4.46 *p* = 0.108 > 0.05], participated in sports clubs [χ^2^ (2) = 1.59 *p* = 0.453 > 0.05] or, sports teams [χ^2^ (2) = 3.30 *p* = 0.193 > 0.05], and played sports with family members or classmates on their own initiative [χ^2^ (2) = 5.48 *p* = 0.065 > 0.05] (Table 2). 

### 3.3. Analysis of Differences in the Extracurricular Activities of Children with Varying Degrees of Fatigue

There was a significant difference between extracurricular activities every day [χ^2^ (2) = 20.89, *p* = 0.001 *p* < 0.001] and the fatigue of young children. Post hoc comparisons revealed that those children with significantly higher degrees of fatigue checked the “almost” daily extracurricular activities on the item of extracurricular activities. In addition, there was no significant difference between those who spent more than one hour on weekdays on extracurricular activities [χ^2^ (2) = 4.36, *p* = 0.113 > 0.05], more than one hour on holidays [χ^2^ (2) = 2.52, *p* = 0.284 > 0.05], and those who had been having extracurricular activities for more than one year [χ^2^ (2) = 1.77, *p* = 0.412 > 0.05] (Table 3).

### 3.4. Analysis of Differences in the Electronic Device Use of Children with Varying Degrees of Fatigue

Watching television or video (smartphones) freely due to boredom [χ^2^ (2) = 6.36, *p* = 0.041 *p* < 0.05], watching television or video (smartphones) on holidays [χ^2^ (2) = 6.30, *p* = 0.043 *p* < 0.05], playing video games or surfing the Internet due to boredom [χ^2^ (2) = 11.79, *p* = 0.003 *p* < 0.01], and playing video games or surfing the Internet on holidays [χ^2^ (2) = 7.02, *p* = 0.030 *p* < 0.05], were significantly different among children’s degrees of fatigue. Post hoc comparisons revealed that the children with significantly higher degrees checked “almost” for watching television or video (smartphones) to pass the time due to boredom, watching television or video (smartphones) on holidays, playing video games, or surfing the Internet due to boredom, and playing video games or surfing the Internet on holidays in the items of electronic device usage. In addition, there was no significant difference between those who spent about one hour a day watching television or video (smartphones) [χ^2^ (2) = 1.47, *p* = 0.479 > 0.05], those who actively asked to watch television or videos (smartphones) [χ^2^ (2) = 2.62, *p* = 0.270 > 0.05], those who spent more than one hour a day playing video games or surfing the Internet [χ^2^ (2) = 2.22, *p* = 0.330 > 0.05], and those who actively asked to play video games or surf the Internet [χ^2^ (2) = 3.12, *p* = 0.210 > 0.05] (Table 4).

## 4. Discussion

This study found that young children who engaged in a variety of appropriate sports, such as exercising at least three times a week, sweating for at least 30 min each time, having a regular exercise habit, and engaging in a variety of sports, were low in fatigue symptoms. In addition, additional sports, such as going on outdoor excursions during holidays, playing outdoors after school, participating in sports clubs and sports teams, and often taking the initiative to ask family members or classmates to play sports with them, were not significantly different. 

This finding is consistent with Kyok and Ikeda’s [19] study of Japanese children. They emphasized that children who had regular outdoor activities had lower fatigue. Bakker et al. [31] assessed 91 patients aged 8 to 18 who completed questionnaires about sleep, somatic symptoms, physical activity, and fatigue. They concluded that poor sleep quality, a physically inactive lifestyle, and specific somatic complaints were important predictive factors. Farmer et al. [32] observed a gradual upward trend in chronic fatigue syndrome (CFS) prevalence in children and adolescents; its prevalence in those over 11 years is equivalent to that in adults. Additionally, according to information on the Harvard Health Publishing website, frequent complaints made by children about being fatigued and the presence of CFS prevent children from enjoying activities and are indicators of health problems [33]. Thus, CFS must not be regarded as affecting adults exclusively. Lee et al. [34] explored the relationship between preschool children’s sleep, exercise abilities, and daily life functions. The results of their study showed that exercise ability had significant effects on daily life functions. Thus, there is a correlation between the appropriateness and diversity of daily activities for children and the manifestation of fatigue symptoms, and developing regular exercise habits in children is an important topic for young children’s health. 

Therefore, it is recommended that parents develop regular exercise habits themselves and encourage children to engage in various types of appropriate activities. The effects of additional activity variables, such as going on outdoor excursions on weekends, playing outdoors after school, participating in sports clubs, sports teams, and often inviting family members or classmates to play sports together, may not be significant. Other factors may be taken into account as well.

This study also found that the degree of fatigue was higher among the children who had extracurricular activities daily. On the other hand, there was no significant difference in the degree of fatigue between children who spent more than one hour on weekdays or holidays on extracurricular activities and those who had been engaged in extracurricular activities for more than one year. 

This result is consistent with the finding of Kyok and Ikeda [19], who pointed out that children spending more than two hours at a time on extracurricular activities had higher degrees of fatigue. In contrast, Shibaki et al. [20] found that there was no significant difference between the degree of fatigue of children and their extracurricular activities. Moreover, Izumi and Maehashi [35] examined the relationship between lessons and the lifestyles of Japanese kindergarten children. They found that 40% of 5~6-year-old children commute to static or dynamic lessons after their regular kindergarten lessons. The children “not taking any lessons” did not spend much time outside the house playing compared to other children who had some lessons. Notably, the amount of time spent watching TV by these children was about two hours a day. Lack of physical activity during the daytime leads to increasingly late bedtimes among those subjects. Children taking swimming or gymnastics dynamic lessons were not relevant to their symptoms of fatigue, which were the same as participating in sports clubs. 

The results of this study imply a correlation between daily extracurricular activities and the degree of fatigue among Taiwanese preschoolers. It is inferred that excessive learning may be associated with fatigue among Taiwanese preschoolers since school ends at 4:00–5:00 and classes of extracurricular activities are held daily. The results of this study do not allow the researchers to determine exactly the difference between the dynamic or statistical status, the frequency and the number of hours each time for children’s extracurricular activities, and the degree of fatigue. Therefore, it is worthwhile to conduct further studies to confirm the differences between the type and duration of extracurricular activities in relation to fatigue among children.

Finally, this study found that the degree of fatigue was higher among children who were allowed to use electronic devices freely due to boredom or during holidays, while there was no significant difference in the degree of fatigue among children who spent one hour per day or who actively requested to use electronic devices. Mitsuoka et al. [23] found that children who used electronic devices for long periods of time, whether during weekdays or holidays, had higher degrees of fatigue. Kyok and Ikeda [19] also suggested that children who watched TV or video for more than two hours had higher degrees of fatigue. 

In addition, Shibaki et al. [21] found that children who watched TV or video for more than three hours had a later bedtime, a shorter sleep time, and higher fatigue. In conclusion, both Taiwanese children who used electronic devices freely to kill time due to boredom or on holidays and Japanese children who used electronic devices for more than two to three hours in a day showed higher degrees of fatigue. Comparing the preschool care time of both countries, Japanese kindergarten care time is mostly half day, while Taiwanese preschool is full day until 4~5 p.m., and extended care is usually provided until 6~7 p.m. to meet the working hours of parents. 

Accordingly, Japanese children use electronic devices for a longer period of time, while Taiwanese children have more time to use electronic devices freely during boredom and holidays; similar factors from the “long time” use of electronic devices and children’s fatigue degree, which are both high. Yang and Chang [36] found that electronic products, such as televisions, smartphones, tablets, and computers, have become an integral part of Taiwanese children’s lives. Among 2164 three-year-old children, only 6.8% had never used electronic products. The electronic products used most frequently by three-year-old children were televisions and smartphones, and the average time children spent watching television was higher than that recommended by the American Academy of Pediatrics. The length of time that children spent using electronic products was linked to their level of development; the longer a child uses these products, the worse their developmental level is. They suggested that parents, caregivers, and educators should pay more attention to the time children spend using these electronic products. 

However, according to Lee [37], parents with young children at home do not know how long a reasonable time to use digital media products should be. Lee and Huang [38] interviewed eight pediatricians and eleven ophthalmologists. The respondents suggested that the (averaged) reasonable stretch of time children spend in front of a screen should be limited to 15 min between ages two and four and 25 min between ages four and six. The (averaged) total amount of screening time each day should not exceed 40 min for children between ages two and four and 60 min for those between ages four and six. The Children Health Handbook [39] suggests that children under two years old should not watch screens; for those older than two years, the time spent watching screens should be no more than one hour a day, and have a 10 min rest after using the eyes for 30 min. 

In addition, there was no significant difference in the degree of fatigue among children who spent one hour per day or actively requested to use electronic devices, probably because the time spent using electronic devices was restricted or they were accompanied by an adult guide. Thus, it was not relevant to the degree of fatigue of the children. On the other hand, the effects of indulgence in electronic devices due to boredom or during holidays on children’s fatigue should not be underestimated. Therefore, it is recommended that parents arrange leisure activities for their children appropriately during the holidays. Moreover, when children use electronic devices, the time spent should be appropriate, and they should be accompanied and guided, which should have a certain degree of effectiveness in preventing children’s fatigue.

## 5. Conclusions and Suggestions

The results of this study show that fatigue among Taiwanese preschoolers is associated with a lack of appropriate, varied, and regular sports activities, excessive extracurricular activities, and prolonged use of electronic devices. Therefore, it is suggested that parents take advantage of the early childhood period being a critical period for habit formation and take a multi-pronged approach in the areas of developing continuous, regular, and appropriate exercise habits. They should also avoid excessive extracurricular activities and the prolonged use of electronic devices. This way, parents may reduce the degree of fatigue and improve the physical health of preschoolers. 

Furthermore, based on the results of this study and the experience of the research process, there are several suggestions for researchers related to this thesis. First, it is suggested that future theories be applied to explore and identify other relevant factors, such as the correlation between parenting stress, parenting behavioral factors, and child fatigue. A limitation of this study is its cross-sectional design, which makes it challenging to establish the temporal sequence of variables and determine the occurrence of fatigue; the causal relationship remains to be analyzed as a reference for follow-up studies. The Children’s Daily Activities Questionnaire only allowed parents to answer “Almost” or “Hardly”. This binary response does not allow the researchers much response variability for subsequent analysis. This is another study limitation. This research has revealed the potential impact of prolonged electronic device usage on young children, leading to fatigue. It highlights the importance of early prevention strategies in addressing these issues.

## Figures and Tables

**Table 1 children-10-01278-t001:** Results of the assessment of CFS in children.

Items	Hardly	Almost
My child seldom engages in physical play	1027 (66.86)	509 (33.14)
My child often yawns already during the day	893 (58.14)	643 (41.86)
My child often looks sleepy	935 (60.87)	601 (39.13)
My child gets becomes tired easily	932 (60.68)	604 (39.32)
My child often becomes irritated	1097 (71.42)	439 (28.58)
My child often has difficulty remaining calm	1044 (67.97)	492 (32.03)
My child often has difficulty concentrating when playing	1087 (70.77)	449 (29.23)
My child often has difficulty sitting still	1019 (66.34)	517 (33.66)
My child often experiences headaches and stomach pain	1126 (73.31)	410 (26.69)
My child often looks unwell	1180 (76.82)	356 (23.18)

**Table 2 children-10-01278-t002:** Chi-square test for homogeneity for the factors influencing sports in children with varying degrees of fatigue.

Item			Group by Degree of Fatigue Number of Times and Percentage			Chi-Square	*p*	Post Hoc Comparison
	Degree		High-Score Group (A)	Middle-Score Group (B)	Low-Score Group (C)			
Exercise at least three times a week	Hardly	Count (ratio in %)	182 (11.85)	319 (20.77)	141	6.05 (2)	0.049 *	B > C
					−9.18			
		Modified residual	−0.03	2	−2.28			
	Almost	Count (ratio in %)	254 (16.54)	398 (25.91)	242			C > B
					−15.76			
		Modified residual	0.03	−2	2.28			
Sweating for at least 30 min at a time	Hardly	Count (ratio in %)	209 (13.61)	353 (22.98)	149	11.38 (2)	0.003 **	B > C
					−9.7			
		Modified residual	0.81	2.17	−3.35			
	Almost	Count (ratio in %)	227 (14.78)	364 (23.70)	234			C > B
					−15.23			
		Modified residual	−0.81	−2.17	3.35			
Going on outdoor excursions during holidays	Hardly	Count (ratio in %)	204 (13.28)	359 (23.37)	163	5.71 (2)	0.058	N.S.
					−10.61			
		Modified residual	−0.24	2.06	−2.13			
	Almost	Count (ratio in %)	232 (15.10)	358 (23.31)	220			
					−14.32			
		Modified residual	0.24	−2.06	2.13			
Playing outdoors after school	Hardly	Count (ratio in %)	307 (19.99)	491 (31.97)	244	4.46 (2)	0.108	N.S.
					−15.89			
		Modified residual	1.36	0.5	−2			
	Almost	Count (ratio in %)	129 (8.40)	226 (14.71)	139 (9.05)			
		Modified residual	−1.36	−0.5	2			
Continuous regular exercise habits	Hardly	Count (ratio in %)	314 (20.44)	483 (31.45)	229	13.94 (2)	0.001 ***	A > C
					−14.91			
		Modified residual	2.74	0.44	−3.36			
	Almost	Count (ratio in %)	122 (7.94)	234 (15.23)	154 (10.03)			C > A
		Modified residual	−2.74	−0.44	3.36			
Variety of exercise types	Hardly	Count (ratio in %)	299 (19.47)	476 (30.99)	226	9.11 (2)	0.011 *	A > C
					−14.71			
		Modified residual	1.97	0.94	−2.92			
	Almost	Count (ratio in %)	137 (8.92)	241 (15.69)	157			C > A
					−10.22			
		Modified residual	−1.97	−0.94	2.92			
Participating in sports clubs	Hardly	Count (ratio in %)	386 (25.13)	648 (42.19)	338	1.59 (2)	0.453	N.S
					−22.01			
		Modified residual	−0.63	1.25	−0.78			
	Almost	Count (ratio in %)	50	69	45			
			−3.26	−4.49	−2.93			
		Modified residual	0.63	−1.25	0.78			
Participating in sports clubsParticipating in sports teams	Hardly	Count (ratio in %)	427 (27.80)	697 (45.38)	367	3.30 (2)	0.193	N.S
					−23.89			
		Modified residual	1.27	0.31	−1.67			
	Almost	Count (ratio in %)	9 (0.59)	20 (1.30)	16			
					−1.04			
		Modified residual	−1.27	−0.31	1.67			
Taking the initiative to invite family members or classmates to play sports together	Hardly	Count (ratio in %)	322 (20.96)	495 (32.23)	255	5.48 (2)	0.065	N.S
					−16.6			
		Modified residual	2.18	−0.6	−1.58			
	Almost	Count (ratio in %)	114 (7.42)	222 (14.45)	128			
					−8.33			
		Modified residual	−2.18	0.6	1.58			

Variables of Significance (* *p* ≤ 0.05, ** *p* ≤ 0.01, *** *p* ≤ 0.001); N.S. = not significant (*p* > 0.05).

**Table 3 children-10-01278-t003:** Chi-square test of homogeneity for the factors influencing extracurricular activities of children with varying degrees of fatigue.

Item			Groups by Degree of Fatigue Number of Times and Percentage			Chi-Square	*p*	Post Hoc Comparison
	Degree		High-Score Group (A)	Middle-Score Group (B)	Low-Score Group (C)			
Spending more than one hour on extracurricular activities on weekdays	Hardly	Count (ratio in %)	273	409	216	4.36 (2)	0.113	N.S.
			−17.77	−26.63	−14.06			
		Modified residual	2.08	−1.06	−0.95			
	Almost	Count (ratio in %)	163	308	167			
			−10.61	−20.05	−10.87			
		Modified residual	−2.08	1.06	0.95			
Spending more than one hour on extracurricular activities on holidays	Hardly	Count (ratio in %)	285	438	247	2.52 (2)	0.284	N.S.
			−18.55	−28.52	−16.08			
		Modified residual	1.13	−1.57	0.63			
	Almost	Count (ratio in %)	151	279	136			
			−9.83	−18.16	−8.85			
		Modified residual	−1.13	1.57	−0.63			
Daily extracurricular activities	Hardly	Count (ratio in %)	332	455	251	20.89 (2)	0.001 **	B > A
			−21.61	−29.62	−16.34			
		Modified residual	−3.23	4.52	−0.99			
	Almost	Count (ratio in %)	104	262	132			A > B
			−6.77	−17.06	−8.59			
		Modified residual	3.23	−4.52	0.99			
Having extracurricular activities for more than one year	Hardly	Count (ratio in %)	157	244	121	1.77 (2)	0.412	N.S.
			−10.22	−15.89	−7.88			
		Modified residual	1.05	0.04	−1.14			
	Almost	Count (ratio in %)	279	473	262			
			−18.16	−30.79	−17.06			
		Modified residual	−1.05	−0.04	1.14			

Variables of Significance (* *p* ≤ 0.05, ** *p* ≤ 0.01); N.S.= not significant (*p* > 0.05).

**Table 4 children-10-01278-t004:** Chi-square test of homogeneity for the factors influencing electronic device use in children with varying degrees of fatigue.

Item			Group by the Degree of Fatigue Number of Times and Percentage			Pearson’s Chi-Square Test	*p*	Post Hoc Comparison
	Degree		High-Score Group (A)	Middle-Score Group (B)	Low-Score Group (C)			
Watching about one hour of TV or video (smartphones) every day	Hardly	Count (ratio in %)	239 (15.56)	414 (26.95)	225 (14.65)	1.47 (2)	0.479	N.S.
		Modified residual	−1.17	0.43	0.72			
	Almost	Count (ratio in %)	197 (12.83)	303 (19.73)	158 (10.29)			
		Modified residual	1.17	−0.43	−0.72			
Requesting to watch TV or video every day (smartphones)	Hardly	Count (ratio in %)	204 (13.28)	369 (24.02)	196 (12.76)	2.62 (2)	0.270	N.S.
		Modified residual	−1.62	1.03	0.5			
	Almost	Count (ratio in %)	232 (15.10)	348 (22.66)	187 (12.17)			
		Modified residual	1.62	−1.03	−0.5			
Watching TV or video (smartphones) freely due to boredom and to kill time	Hardly	Count (ratio in %)	239 (15.56)	433 (28.19)	238 (15.49)	6.36 (2)	0.041 *	C > A
		Modified residual	−2.22	0.86	1.97			
	Almost	Count (ratio in %)	197 (12.83)	284 (18.49)	145 (9.44)			A > C
		Modified residual	2.22	−0.86	−1.97			
Watching TV or video (smartphones) freely on holidays	Hardly	Count (ratio in %)	221 (14.39)	401 (26.11)	227 (14.78)	6.30 (2) *	0.043 *	C > A
		Modified residual	−2.28	0.48	1.97			
	Almost	Count (ratio in %)	215 (14.00)	316 (20.57)	156 (10.16)			A > C
		Modified residual	2.28	−0.48	−1.97			
Playing video games or surfing the Internet for about one hour every day	Hardly	Count (ratio in %)	302 (19.66)	507 (33.01)	283 (18.42)	2.22 (2)	0.330	N.S.
		Modified residual	−0.99	−0.31	1.39			
	Almost	Count (ratio in %)	134 (8.72)	210 (13.67)	100 (6.51)			
		Modified residual	0.99	0.31	−1.39			
Requesting to play video games or surf the Internet every day	Hardly	Count (ratio in %)	286 (18.62)	484 (31.51)	273 (17.77)	3.12 (2)	0.210	N.S.
		Modified residual	−1.22	−0.31	1.63			
	Almost	Count (ratio in %)	150 (9.77)	233 (15.17)	110 (7.16)			
		Modified residual	1.22	0.31	−1.63			
Playing video games or surfing the Internet freely due to boredom or to kill time	Hardly	Count (ratio in %)	282 (18.36)	497 (32.36)	290 (18.88)	11.79 (2)	0.003 **	C > A
		Modified residual	−2.64	−0.22	3.01			
	Almost	Count (ratio in %)	154 (10.03)	220 (14.32)	93 (6.05)			A > C
		Modified residual	2.64	0.22	−3.01			
Playing video games or surfing the Internet freely on holidays	Hardly	Count (ratio in %)	290 (18.88)	499 (32.49)	287 (18.68)	7.02 (2)	0.030 *	C > A
		Modified residual	−1.91	−0.37	2.41			
	Almost	Count (ratio in %)	146 (9.51)	218 (14.19)	96 * (6.25)			A > C
		Modified residual	1.91	0.37	−2.41			

Variables of Significance (* *p* ≤ 0.05, ** *p* ≤ 0.01; N.S. = not significant (*p* > 0.05).

## Data Availability

Not applicable.

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
