# Peer review of "Study on Sports, Extracurricular Activities, Electronic Device Usage Factors Associated with Chronic Fatigue Syndrome in Taiwanese Preschoolers"

_children, 2023, doi:10.3390/children10081278_

Round 1
Reviewer 1 Report
Dear Authors,
I have received an article which assessed the effects of sports, extra-curricular activities, and electronic device usage factors on Chronic Fatigue Syndrome (CFS) in Taiwanese Preschoolers. This manuscript seems to be a larger project for dissertation which has been published in Children too (https://www.mdpi.com/2227-9067/10/7/1149). As for this specific manuscript, I have several points that need to be addressed:
1) Please provide the numerical p-value and effect sizes in the abstract
2) The introduction is too lengthy and incoherent. Some parts are more suitable for the discussion section.
3) Line 47 --> Did Yu (2022) really studied health posture (such as slouching or sitting straight) or did the authors mean something else?
4) Lines 139-140 are more suitable for the results section
5) Any attempts on proper translation from the Japanese questionnaire to the Chinese one? If not, this should be included in the limitation section
6) There is a lack of description on data analyses. In the previous manuscript the authors described how post-hoc analyses were conducted, but this is missing in this manuscript
7) Line 263 --> What is a homogeneity test?
8) Please put the exact p-value in the results section, just like how the authors delineated them in the tables.
9) Please include a limitation section
The English needs major improvement as it is hard for me to read this manuscript
Reviewer 2 Report
This research aimed to investigate the sports, extracurricular activities, and electronic device usage factors associated with chronic fatigue syndrome among Taiwanese preschoolers. Five year-old preschoolers were randomly selected using a stratified multi-stage random cluster sampling method. The parents of the preschoolers completed the questionnaires containing items related to the symptom of fatigue, extracurricular activities, and the electronic device usage of their preschoolers.
The study is very interesting and is well structured in every part. I have only one minor suggestion for the authors. In the introductory paragraph they may refer to some more recently published work. In this regard, the authors could take inspiration from a recent work that analyzed the lifestye conditions of students using a questionnaire (Moscatelli et al., Assessment of Lifestyle, Eating Habits and the Effect of Nutritional Education among Undergraduate Students in Southern Italy. Nutrients 2023, 15(13), 2894)
Check all refernces and refereces list.
Author Response
Thank you for the reviewer's suggestions. We have made the necessary revisions in lines 79-81.
Reviewer 3 Report
Dear Authors
I was given a manuscript to check titled: Study on Sports, Extracurricular Activities, Electronic Device Usage Factors Associated with Chronic Fatigue Syndrome in Taiwanese Preschoolers.
From my point of view, this paper contains some methodological gap that affect the quality of the article against a subject of the research exceedingly actual and important.
Therefore, I indicate some recommendations for improvements in the work presented.
Suggested improvements are:
Abstract. The authors should start with a short intro that better highlights their work and end up with a paragraph that include results.
A lot of necessary information is missing in methods section:
- More information should be provided about the participants’ characteristics
- What were inclusion and exclusion criteria?
- Is there an experimental procedure? It is not discussed.
- What was done to improve the adherence to the participants' intervention?
The discussion could benefit from further information regarding the existing theory. In addition, the discussion does not address the limitations and strength of the study and it is necessary that it start with a first paragraph describing the main aims and then the main results.
Best Regards
Round 2
Reviewer 1 Report
All suggestions have been accepted and implemented in the manuscript. I have no further comments.
Minor editing is required (Some sentences still feel directly translated from the mother tongue)